# *FGF9*-Associated Multiple Synostoses Syndrome Type 3 in a Multigenerational Family

**DOI:** 10.3390/genes14030724

**Published:** 2023-03-15

**Authors:** Ariane Schmetz, Jörg Schaper, Simon Thelen, Majeed Rana, Thomas Klenzner, Katharina Schaumann, Jasmin Beygo, Harald Surowy, Hermann-Josef Lüdecke, Dagmar Wieczorek

**Affiliations:** 1Institute of Human Genetics, Medical Faculty and University Hospital Düsseldorf, Heinrich Heine University Düsseldorf, 40225 Düsseldorf, Germany; 2Center for Rare Diseases, Medical Faculty and University Hospital Düsseldorf, Heinrich Heine University Düsseldorf, 40225 Düsseldorf, Germany; 3Department of Orthopedic and Trauma Surgery, Medical Faculty and University Hospital Düsseldorf, Heinrich Heine University Düsseldorf, 40225 Düsseldorf, Germany; 4Department of Oral, Maxillo- and Plastic Facial Surgery, Medical Faculty and University Hospital Düsseldorf, Heinrich Heine University Düsseldorf, 40225 Düsseldorf, Germany; 5Department of Otorhinolaryngology, Medical Faculty and University Hospital Düsseldorf, Heinrich Heine University Düsseldorf, 40225 Düsseldorf, Germany; 6Institute of Human Genetics, University Hospital Essen, University Duisburg-Essen, 45147 Essen, Germany

**Keywords:** *FGF9*, multiple synostoses syndrome type 3, SYNS3, cleft palate, craniosynostoses, fusion of interphalangeal joints, whole exome sequencing, human genetics

## Abstract

Multiple synostoses syndrome (OMIM: #186500, #610017, #612961, #617898) is a genetically heterogeneous group of autosomal dominant diseases characterized by abnormal bone unions. The joint fusions frequently involve the hands, feet, elbows or vertebrae. Pathogenic variants in *FGF9* have been associated with multiple synostoses syndrome type 3 (SYNS3). So far, only five different missense variants in *FGF9* that cause SYNS3 have been reported in 18 affected individuals. Unlike other multiple synostoses syndromes, conductive hearing loss has not been reported in SYNS3. In this report, we describe the clinical and selected radiological findings in a large multigenerational family with a novel missense variant in *FGF9*: c.430T>C, p.(Trp144Arg). We extend the phenotypic spectrum of SYNS3 by suggesting that cleft palate and conductive hearing loss are part of the syndrome and highlight the high degree of intrafamilial phenotypic variability. These findings should be considered when counseling affected individuals.

## 1. Introduction

Multiple synostoses syndrome is a group of autosomal dominant entities characterized by abnormal bone unions. Joint fusions frequently affect the hands, feet, elbows or vertebrae. Pathogenic variants in *NOG*, *GDF5*, *FGF9* and *GDF6* have been associated, respectively, with the four molecular subtypes of multiple synostoses syndromes (SYNS1-4). In addition to bone fusions, characteristic facial features as well as progressive conductive hearing loss due to otosclerosis and stapes fusion have been described in patients with SYNS1 [1,2] and SYNS4 [3]. Therefore, SYNS1 has also been referred to as facio-audio-symphalangism syndrome in the past [4].

SYNS3 is caused by heterozygous missense variants in the fibroblast growth factor 9 (*FGF9*) gene. Mammalian fibroblast growth factors (FGFs) are a family of secreted proteins that exert pleiotropic effects by binding with four signalling tyrosine kinase FGF receptors (FGFRs). Activated FGFRs transduce the signal through downstream intracellular signalling pathways, mainly RAS-MAPK, PI3K-AKT, PLCγ and STAT [5]. This FGF signalling plays an essential role during embryonic development and organogenesis and in adult tissues. The important role of FGF signalling in coordination of the development of bones and joints is highlighted by its implications in diseases [6]. Thus, pathogenic variants in the genes encoding FGFR1-3 are associated, among others, with osteochondrodysplasia and craniosynostoses.

In the spontaneous dominant mouse mutant *Elbow knee synostosis* (*Eks*), the Asn143Thr missense mutation in the *fgf9* gene has been shown to impair *FGF9* protein homodimerization, reduce the affinity to heparin and possibly increase diffusion of the altered *FGF9* protein, resulting in ectopic *FGF9* signalling [6]. As a consequence of altered FGF signalling, the mutant mice present elbow joint fusion and knee joint dysplasia as constant traits in heterozygotes and shortened limbs and tails, vertebral and thoracic alterations, and premature cranial sutures fusions and clefts in the secondary palate in homozygotes [7]. It has been shown that *fgf9* inhibits osteogenesis and promotes oesteoclastogenesis [8]. Therefore, *fgf9* is a well-known factor that regulates bone development, but its exact function in bone homeostasis is still under investigation. However, the most likely disease-causing mechanism is a dominant negative effect of missense variants in multimeric complex rather than haploinsufficiency. This assumption is further supported by reports about heterozygous deletion encompassing *FGF9* in humans who do not present with a SYNS3 phenotype [9].

To date, to the best of our knowledge, only five heterozygous disease-causing missense *FGF9* variants have been described in 18 affected individuals with SYNS3 from six families [10,11,12,13,14,15].

Here, we report on a multigenerational family with a total of 29 affected individuals, who display some previously unknown features such as cleft palate and conductive hearing impairment, with a novel, likely pathogenic, heterozygous missense variant in *FGF9*. Herein, we extend the phenotypic spectrum of SYNS3 and demonstrate the intrafamilial phenotypic variability.

## 2. Materials and Methods

We describe a large five-generational family with autosomal-dominant inherited skeletal anomalies (Figure 1). We performed a physical examination on one patient (patient VI.1.), his dizygotic twin brother (patient VI.2.) and their father (patient V.1.). Audiometric testing was performed on patients VI.2. and V.1. The audiologic assessment included pure-tone audiometry and tympanometry. Pure-tone audiometry was performed according to the ISO guidelines (ISO 8253-1:2010). Air conduction as well as bone conduction thresholds were assessed. Before audiometry, a microscopic otoscopy was performed. No radiographs were available for patient VI.1. Radiographs of the feet, hands and left elbow, a computed tomography of the left elbow, and MR imaging of the lumbar spine were available for patient VI.2. Ancient radiographs of the elbow and a computer tomography of the spine were available for patient V.1. A written consent for publication of anonymized data and photographs was obtained from the three patients. Additionally, extensive clinical and radiological information on another 26 affected family members was available [16].

Genomic DNA was extracted from peripheral blood leucocytes by standard procedures for the three patients. Whole exome sequencing was performed in patients VI.1. and V.1. using the Twist Biosciences Library Preparation EF and Standard Hybridization and Wash kits for library preparation, together with the Human Comprehensive Exome panel for target enrichment. Sequencing was performed on an Illumina NextSeq 550 DNA sequencer with a NextSeq550 High-Output kit v.2.5. The varvis^®^ genomics platform (Limbus Medical Technologies, Rostock, Germany) was used for a bioinformatic analysis of the sequence raw data and variant identification based on the human reference genome GRCh38, as well as for subsequent data interpretation. The priorization of variants of the duo-exome data was done by filtering for autosomal dominant inheritance (0/1 in father and son), allele frequency in the control population (gnomAD) and HPO similarity using the varvis^®^ HPOSimScore with the terms cleft palate (HP:0000175), humeroradial synostosis (HP:0003041) and sagittal craniosynostosis (HP:0004442). Sanger sequencing was performed for verification in patients VI.1. and V.1. and for testing in patient VI.2. and the unaffected sister (VI.3.).

The zygosity of the twin brothers was tested using the PowerPlex16 HS System (Promega, Madison, WI, USA), which tests 15 short tandem repeat markers simultaneously and in addition the *Amelogenin* gene to discriminate the gonosomes. The amplified PCR products were analyzed on a 3130xl Genetic Analyzer (Applied Biosystems, Foster City, CA, USA), and the GeneMarker Software (SoftGenetics, State College, PA, USA) was used for data analysis.

## 3. Results

### 3.1. Clinical Data

A 25-year-old man (patient VI.1.) was referred to our clinical genetics department with suspected syndromic craniosynostosis. He had a history of surgically treated sagittal craniosynostosis and cleft palate. During childhood, he underwent surgical correction for the reduced mobility of his thumbs. Extension deficits in his elbows were known since early childhood. Physical examination revealed an almost complete ankylosis of both elbow joints in 70° of flexion and neutral pronation (Figure 2a). He also had a pectus carinatum and bilateral absence of skin creases over his distal interphalangeal joints, with reduced mobility of his distal interphalangeal joints, suggesting a synostosis of the middle and distal phalanges (Figure 2b). Radiographs were not available.

His dizygotic twin brother (patient VI.2.) presented with a slightly limited range of motion of the elbow joints. Pronation was limited by 10°, and extension of the elbow was limited by 30° (Ext/Flex 0-30-140). The range of motion of his thumbs also appeared to be slightly limited. His halluces were broad and showed valgus deviation (Figure 3a). In childhood, a bilateral hallux valgus deformity was surgically corrected; and in adolescence, a calcaneonavicular coalition was surgically corrected on both feet. He also has pectus carinatum and bilateral absence of skin creases over the distal interphalangeal joints, and he reported lumbar back pain.

Radiographs of the left elbow showed a dysplastic distal humerus with a tiny trochlea and dysplastic olecranon and a dysplastic convex radial head (Figure 3b). A computed tomography of the elbow depicted the malformation (Figure 3c). Radiographs of the hands displayed a dysplastic first metacarpal base and head, flattening of the heads of metacarpals 3 and 4 as well as synostosis of the middle and distal phalanges of the fifth fingers (Figure 3d). The radiographs of the feet before correction of the calcaneonavicular coalition displayed a coalition between the anterior processus calcanei and the os naviculare. An MRI of the spine showed partial lumbar vertebral synostosis L3-L4 (Figure 3e).

An audiometric evaluation showed normal hearing on the left side. On the right side, a mild conductive hearing loss of 20 dB in the low and middle frequencies and 30 dB in the higher frequencies was measured (Appendix A). The bone conduction threshold showed a normal inner ear with hearing levels at 0–10 dB, except for a slightly increased bone conduction threshold at 2 kHz to 20 dB, which is similar to Carhart notch, indicative of otosclerosis. A tympanometry showed a type A tympanogram on both sides, indicating a good compliance of the tympanic membrane. Stapedius reflexes could not be detected ipsi- or contralaterally on the right side. An otoscopy showed no pathologies.

The father (patient V.1.) underwent thumb surgery in childhood and reported a lumbar synostosis. He has a submucous cleft palate. The range of motion of his elbows is limited, with almost complete ankylosis in 45° of flexion and inability to pronate (Figure 4a). He has broad hallux valgus bilaterally, and a hearing impairment is known.

Radiographs of his elbows at 3 (Figure 4b) and 6 (Figure 4c) years of age show complete humeroradial synostosis on the right side and dysplastic distal humerus, with tiny trochlea and malformed olecranon and radial head with developing humeroradial and humeroulnar synostosis, on the left side. There is evidence of the progression of the synostosis in childhood.

An audiometric evaluation revealed a moderate mixed hearing impairment on the right side as well as a profound mixed hearing impairment on the left side (Appendix A). On the right side, an air conduction threshold of 50–60 dB HL was measured. There was an air-bone gap of 30–40 dB at 250 Hz and 500 Hz. On the left side, there was an air conduction threshold of 70–90 dB HL, with an air-bone gap of 40–50 dB in the lower frequencies, 30 dB in the middle frequencies, 5 dB at 2 kHz and 30–40 dB from 2–4 kHz. A tympanometry showed a type B pattern on the left side and a type C pattern on the right side. The stapedius reflex could only be detected on the right ipsilateral side. An otoscopy revealed an intact tympanic membrane reconstruction with a radical cavity on the left side and a slightly retracted tympanic membrane on the right side.

Progression of the skeletal symptoms was not reported for any of the patients. All affected family members have normal intelligence.

Family history included skeletal anomalies in five generations (Figure 1). The family was first described in 1930. In 1973, a comprehensive clinical and radiological work-up of this family was performed as part of a medical doctoral thesis [16]. Patient V.1. was also included in this former work and was two-and-a-half years old at that time. In the extensive radiological examinations, all types of joint anomalies from narrow joint gaps to partial and complete synostosis of joints could be identified. Mostly, the anomalies were symmetrical; however, the severity varied between both sides.

The constant features in all 27 subjects were the elbow anomalies with deformities of the cubital joint. Symphalangy of toes and fingers, tarsal synostoses and brachybasophalangy and brachymesophalangy of feet and hands were common features.

Two patients (individuals V.4. and V.16. in Figure 1) were incidentally found to have vertebrae synostosis. No systematic radiological examinations of the spine were performed for the remaining subjects. A cleft palate was described in one patient (individual V.9. in Figure 1). However, the covered cleft palate in patient V.1. was not noticed in the original examination. The clinical examinations do not mention any craniosynostoses, but at least one patient (individual III.4. in Figure 1) shows facial asymmetry on family pictures.

### 3.2. Genetic Testing Results

Whole exome sequencing (WES) in patient VI.1. and his father patient V.1. identified the heterozygous missense variant NM_002010.2:c.430T>C p.(Trp144Arg) in Exon 3 of the *FGF9* gene. Missense variants in *FGF9* have been associated with the autosomal dominant multiple synostoses syndrome 3 (SYNS3). The presence of the variant in the exome-sequenced patients and the dizygous twin brother (patient VI.2.) as well as its absence in the unaffected sister were confirmed with Sanger sequencing (Figure 5a).

This variant leads to the substitution of a tryptophan residue for arginine at position 144. Trp144, at the top of the dimer interface of the protein, is involved in the formation and stabilization of the mainly hydrophobic contacts between the monomer components of the *FGF9* dimer [17].

This domain of *FGF9* is conserved down to *Xenopus* [18] (Figure 5b). The variant is absent from the general population (gnomAD database v. 3.1.2 and 2.2.1 accessed 01/2023 [19] and GoNL SNPs and Indels release 5 accessed 02/2023 [20]) and is not listed in the ClinVar database [21]. Multiple in silico predictors (MutationTaster, PROVEAN, CADD, GERP++) point towards a pathogenic effect of this variant.

**Figure 5 genes-14-00724-f005:**
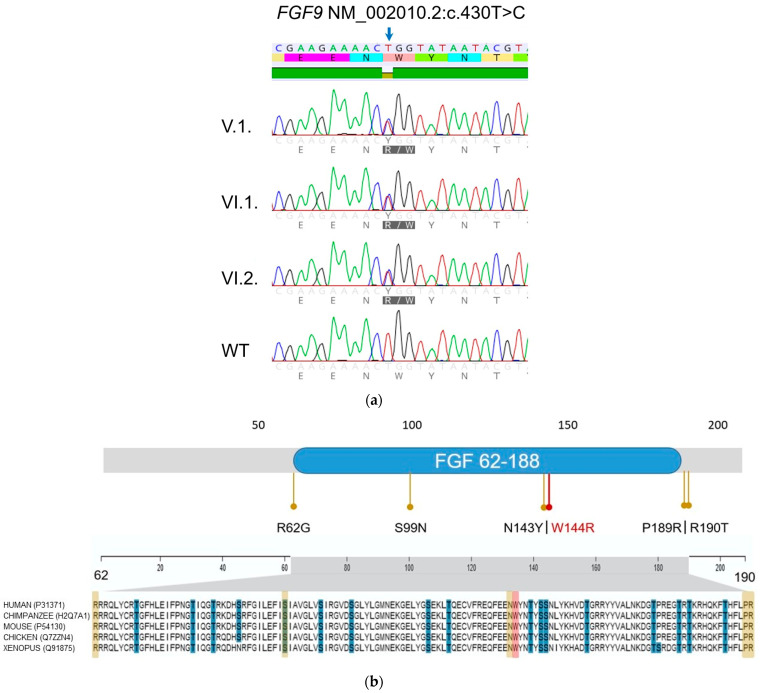
(**a**) Sanger chromatogram of the *FGF* variant c.430T>C in patients VI.1., VI.2., V.1. and wild-type allele (WT). (**b**) Upper-row protein domain structure of *FGF9*, with the known missense variants causing SYNS3 marked by yellow dots and the variant in this report (W144R) marked in red. Lower-row protein sequence alignment down to Xenopus; note the highly conserved amino acid sequence (this figure was partially build upon material from UniProt [22] licenced under CC BY 4.0 (https://creativecommons.org/licenses/by/4.0/)).

## 4. Discussion

Here, we describe the seventh family with *FGF9*-associated multiple synostoses syndrome. The detailed clinical and radiological data from 29 affected individuals from a multigenerational family spanning five generations impressively highlight the intrafamilial variability, while suggesting complete penetrance. Our data, together with data from the literature, are summarized in Table 1.

Wu et al., (2009) identified heterozygous variants in *FGF9* as one of the causes of SYNS (SYNS3) by linkage analysis in a five-generation family with 12 affected individuals [10]. In the described family, anomalies of the elbow joint were a constant feature, synostosis of hands and feet occurred in more than one half of the individuals and 25% of individuals had lumbar synostosis. Neither cleft palate nor craniosynostoses or hearing impairment were reported.

Subsequently, Rodriguez-Zabala et al., (2017) described a father and son with *FGF9*-associated multiple synostoses. In addition to the anomalies of the hands and feet, both affected individuals had craniosynostosis. The father, in whom the variant in *FGF9* occurred *de novo*, was also described with vertebral fusion and cleft palate [11]. The same variant was also identified in a multiplex targeted high-throughput sequencing effort in one patient out of a cohort of patients with congenital limb malformations [12].

In 2020, Sentchordi-Montané et al. and Thuresson et al. described a case of an 11-year-old girl and a 16-year-old boy, respectively, with *FGF9*-associated multiple synostoses syndrome [13,14]. The boy had a high narrow palate with bifid uvula.

Most recently, Dobson et al., (2022) described the first patient with *FGF9*-associated multiple synostoses syndrome with learning disability [15].

Taken together, the most constant feature is the anomaly of elbow joint, which occurred in 94% of all patients. This feature is absent only in the two patients described by Rodriguez-Zabala et al. and in the patient described by Sentchordi-Montané et al. [11,13]. Vertebrae synostoses and anomalies of the hands and feet occurred in 20–64% of all patients. However, the number of vertebrae synostosis could be underestimated since no systematic radiological examination of the spine was performed.

With this report, we confirm the association of craniosynostoses with pathogenic variants in *FGF9* by describing the second family with at least one affected individual with craniosynostosis associated with SYNS3, after that of Rodriguez-Zabala et al. The authors performed extensive analyses, including proximity ligation assays, to compare the degree of homodimerization between the wild type and *FGF9* mutations and concluded that the mutation in *FGF9* is responsible for the craniosynostoses [11].

The variant detected in this family is located at the top of the dimer interface, and only one amino acid downstream of the p.(Asn143Tyr) variant was identified in a girl [13] at the same position as the well-characterized spontaneous elbow knee synostoses (*Eks*) mouse mutant variant p.(Asn143Thr). This variant is known to cause elbow joint fusion and knee joint dysplasia in heterozygous mice and premature fusions of coronal and sagittal sutures and a cleft in the secondary palate in homozygous mice [6,7].

Prior to this report, a cleft palate has only been described once in a human with SYNS3 [11]; however, the authors of this study did not discuss the cleft palate in relation to the *FGF9* pathogenic variant. In this family, three individuals had clefting. However, the covered cleft palate in patient V.1. was not noticed in the original examination, indicating that this feature could be underestimated. Additionally, the bifid uvula in one individual of a previous report [14] could be seen as the mildest form of a cleft palate. We postulate that cleft palate must be considered as part of the clinical spectrum of SYNS3, since murine *fgf9* has been shown to participate in palatogenesis [23].

Hearing impairment has not been described in relation to SYNS3 so far, albeit it is a common feature in SYNS1 and SYNS4. In this report, patient VI.2. had audiologic testing results that were suggestive of otosclerosis. Patient V.1. had mixed hearing impairment. On the one hand, within patients with otosclerosis, about one third develop a clinically relevant sensorineural component of hearing impairment, which then manifests as a mixed hearing impairment [24], similarly to that of patient V.1. On the other hand, studies have demonstrated an important role of *FGF9* in cochlea formation in mice [25], so that an effect of *FGF9* missense variants in humans on sensory hearing impairment cannot be excluded either. Although a definite association between SYNS3 and hearing impairment cannot be conclusively established through these isolated cases, we would recommend audiometric testing in patients with SYNS3 to maintain the possibility of early intervention.

Several previous reports [10,11,13] have postulated an age-dependent effect and progression of SYNS3, as has been illustrated for SYNS1 [4]. This family’s history partially corroborates this assumption. The radiologic findings of patient V.1., aged 3 and 6 years, showed progression during childhood. However, none of the patients reported progression of the skeletal symptomatology.

## 5. Conclusions

With this report on a large family with SYNS3, we highlight the intrafamilial variability of this highly penetrant disorder. We confirm in a second family that craniosynostoses and cleft palate, although less common, are part of the phenotypic spectrum of SYNS3. In addition, the studied family indicates that conductive hearing impairment may also be associated with SYNS3.

## Figures and Tables

**Figure 1 genes-14-00724-f001:**
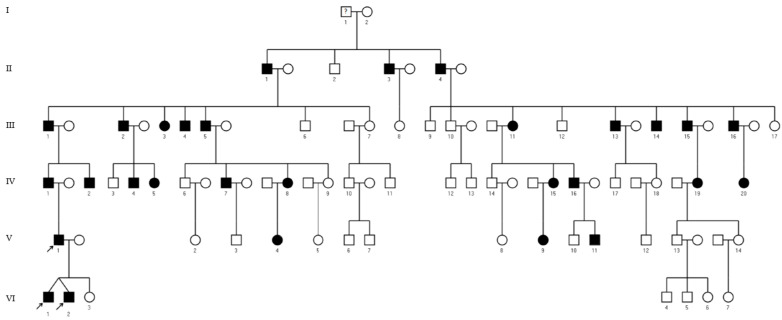
Pedigree with autosomal-dominant SYNS3. Squares and circles indicate males and females, respectively. Roman numerals designate the different generation of the family and latin numerals the individuals in a generation. Affected individuals are indicated by black symbols. Arrows mark the patients described in more detail in this report: patients VI.1., VI.2. and V.1. Information concerning the deceased was not available.

**Figure 2 genes-14-00724-f002:**
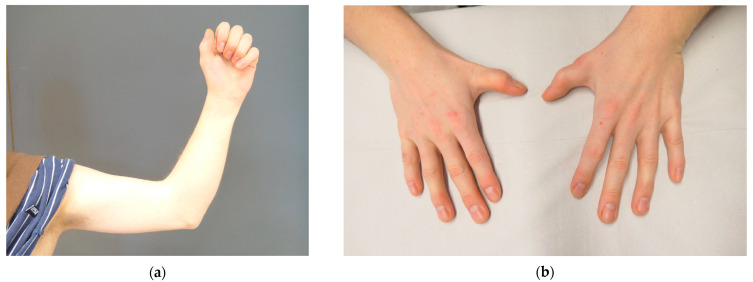
Photographs of patient VI.1. (**a**) Left elbow: the patient has almost complete absence of mobility of the elbow in flexion and extension and for pronation and supination. (**b**) Note the shortened fifth fingers, the missing creases over the distal interphalangeal joints of the fifth fingers and the shortened distal phalanx of the thumbs.

**Figure 3 genes-14-00724-f003:**
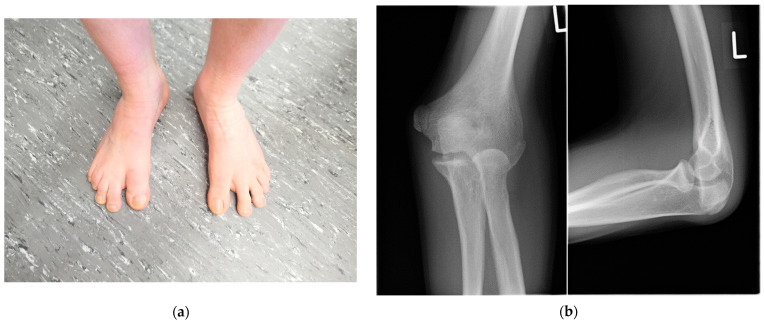
Patient VI.2. (**a**) Photograph of his feet after surgical correction of hallux valgus: note the broad and still valgus deviated halluces and sandal gaps. (**b**) The radiographs of the left elbow showed a dysplastic distal humerus with a tiny trochlea and dysplastic olecranon and a dysplastic convex radial head. (**c**) A computed tomography of the elbow depicted the malformation (upper row: coronal view, lower row: sagittal view). (**d**) Radiographs of the hands displayed a dysplastic first metacarpal base and head, flattening of the heads of metacarpal 3 and 4 as well as synostosis of the middle and distal phalanges of the fifth fingers. (**e**) An MRI of the lumbar spine revealed a partial posterior vertebral synostosis marked by arrows of L3 and L4 (upper row: T1-weighted images, lower row: T2-weighted images).

**Figure 4 genes-14-00724-f004:**
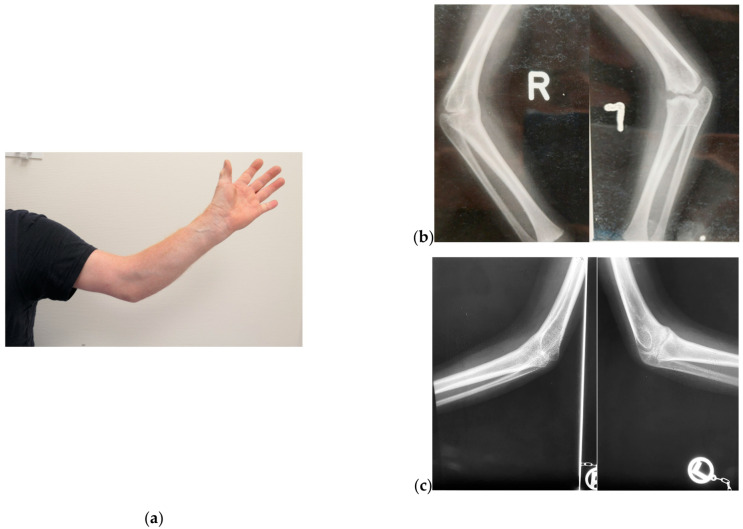
Photographs of patient V.1. (**a**) Left elbow: the patient has no range of motion of the elbow in flexion and extension and for pronation and supination. He is fixed in 45° of flexion and pronation (**b**) Radiograph of the elbows at 3 years of age (Courtesy Ingo Rehmann with permission) and (**c**) 6 years of age. There is progressive humeroradial synostosis on the right side and developing humeroradial and humeroulnar synostosis on the left side.

**Table 1 genes-14-00724-t001:** Summary of the phenotypic traits seen in the described patients, the studied pedigree and the available literature.

	Patient 1	Patient 2	Patient 3	Studied Pedigree	Wu et al., (2009) [10]	Rodriguez-Zabala et al., (2017) [11]	Jourdain et al., (2020) [12]	Sentchordi-Montané et al., (2020) [13]	Thuresson et al., (2021) [14]	Dobson et al., (2022) [15]	Total
*FGF9* (NM_002010.2)	c.430T>C, p.(Trp144Arg)				c.296G>A, p.(Ser99Asn)	c.184A>G, p.(Arg62Gly)		c.427A>T, p.(Asn143Tyr)	c.566C>G, p.(Pro189Arg)	c.569G>C p.(Arg190Thr)	6 missense variants
Female/male	0/1	0/1	0/1	9/17	3/9	0/2	1 (ND)	1/0	0/1	1/0	47
Craniosynostosis	+	-	-	0/26 *	0/12 *	2/2	ND	0/1	0/1	0/1	3/46 (7%)
Cleft palate	+	-	+	1/26 *	0/12 *	1/2	ND	0/1	0/1	0/1	4/46 (9%)
Anomaly of elbow joint	+	+	+	26/26	12/12	0/2	1/1	0/1	1/1	1/1	44/47 (94%)
Anomaly of thumbs	+	+	+	11/26	10/12	2/2	1/1	0/1	1/1	1/1	30/47 (64%)
Synostosis of digits II-V	+	+	-	11/26		1/2	ND	1/1	0/1	0/1	
Synostosis of feet	-	+	-	9/26	8/12	1/2	1/1	1/1	1/1	1/1	23/47 (49%)
Vertebrae synostosis	-	+	+	2/26	3/12	1/2	ND	0/1	1/1	0/1	9/46 (20%)
Others	Crowded teeth	Hearing impairment	Hearing impairment					Back pain	Bifid uvula	Learning disability	
		Back pain							Crowded teeth		

Abbreviations: + present, - absent, and ND: not described. * No explicit data for all patients was available for these items, but in the context of the extensive clinical workup done in the concerned studies, we suppose that those phenotypic traits would have been noted and reported.

## Data Availability

The data that support the findings of this report are openly available in ClinVar (https://www.ncbi.nlm.nih.gov/clinvar/), reference number: SCV003761540. Further data that support the findings of this study are available on request from the corresponding author. These data are not publicly available due to privacy or ethical restrictions.

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
