# Peer review of "FGF9-Associated Multiple Synostoses Syndrome Type 3 in a Multigenerational Family"

_genes, 2023, doi:10.3390/genes14030724_

Round 1

Reviewer 1 Report

Comments and suggestions for the author

Schmetz et al. identified a novel missense variant in FGF9 and expanded the phenotypic spectrum of synostoses syndrome type 3 (SYNS3) by suggesting cleft palate and conductive hearing loss. Here are few suggestions for author.

·      Author claimed the hearing impairment as additional phenotype for SYNS3, there are few studies which support the expression of FGF9 role in inner ear e.g, https://doi.org/10.1016/j.ydbio.2004.06.010. So the role of FGF9 in inner ear need to mention in the discussion.

·      A figure showing the Sanger chromatogram also which shows the location of all the identified variants in various domains of FGF9 for SYNS3.

·      Please enlist the variants which were prioritized after exome analysis. This can be added in the supplementary table

·      What id CADD and GERP++  score of the variant c.430T>C: p.(Trp144Arg)?

·      Please add a diagram to show the conservation of the variant among various species.

·      Please add a

·      The protein modeling should be performed for the variant c.430T>C: p.(Trp144Arg). Which shows how this variant affects the protein domain?

·      Author didn’t check the variant frequency in bravo, GME, and all of us databases? Please check and update it in manuscript

·      Which other variants were tested by Sanger Sequencing, other then the variant segregated in Family?

·      Mention other phenotypes reported with FGF9 such as seizures https://www.nature.com/articles/s41419-021-03478-1. Mention whether the role of the gene in the nervous system is linked to hearing loss phenotype.

·      Briefly mention the criteria to prioritize the variant in the FGF9 gene, for example, allele frequency, pathogenicity scores, etc.

Author Response

Dear reviewer,

thank you for the evaluation of our manuscript. We are grateful for your pertinent suggestions. Please see our point per point reply attached.

Kind regards, 

Ariane Schmetz and colleagues

Reviewer 2 Report

The authors present a large five generational family with autosomal dominant inherited FGF9-associated Multiple Synostoses Syndrome Type 3 including 29 affected family members which is a rarity per se. The aricle is well written in all parts, the photos have a good quality and the discussion is sound. The table is indeed informative and is summing up all details in literature. Is there a radiograph of the skull of patient VI.1 before surgical treatment?

Author Response

Dear reviewer,

thank you for the very kind evaluation of our manuscript. Unfortunately, solely a CT-Scan had been performed. We were able to get the images but only in a printed version and sadly the quality is too poor for publication (please see PDF attached)

Kind regards, 

Ariane Schmetz and colleagues

Round 2

Reviewer 1 Report

The author made the necessary changes as indicated.